# The Role of Chaperone-Mediated Autophagy in Tissue Homeostasis and Disease Pathogenesis

**DOI:** 10.3390/biomedicines12020257

**Published:** 2024-01-23

**Authors:** Rut Valdor, Marta Martinez-Vicente

**Affiliations:** 1Immunology-Cell Therapy and Hematopoietic Transplant Group, Department of Biochemistry and Molecular Biology B, University of Murcia (UMU), 30100 Murcia, Spain; 2Unit of Autophagy, Immune Response and Tolerance in Pathologic Processes, Biomedical Research Institute of Murcia-Pascual Parrilla (IMIB), 30120 Murcia, Spain; 3Autophagy and Lysosomal Dysfunction Lab, Neurodegenerative Diseases Research Group, Vall d’Hebron Research Institute—CIBERNED, 08035 Barcelona, Spain

**Keywords:** chaperone-mediated autophagy, homeostasis, pathogenesis, disease

## Abstract

Chaperone-mediated autophagy (CMA) is a selective proteolytic pathway in the lysosomes. Proteins are recognized one by one through the detection of a KFERQ motif or, at least, a KFERQ-like motif, by a heat shock cognate protein 70 (Hsc70), a molecular chaperone. CMA substrates are recognized and delivered to a lysosomal CMA receptor, lysosome-associated membrane protein 2A (LAMP-2A), the only limiting component of this pathway, and transported to the lysosomal lumen with the help of another resident chaperone HSp90. Since approximately 75% of proteins are reported to have canonical, phosphorylation-generated, or acetylation-generated KFERQ motifs, CMA maintains intracellular protein homeostasis and regulates specific functions in the cells in different tissues. CMA also regulates physiologic functions in different organs, and is then implicated in disease pathogenesis related to aging, cancer, and the central nervous and immune systems. In this minireview, we have summarized the most important findings on the role of CMA in tissue homeostasis and disease pathogenesis, updating the recent advances for this Special Issue.

## 1. Introduction

In recent years, there have been many works that highlight chaperone-mediated autophagy (CMA) as a mechanism of lysosomal protein degradation that is key to maintaining the homeostasis and different specific functions in the cells of different tissues.

CMA regulates physiologic functions in different organs and is then implicated in disease pathogenesis related to aging, cancer, and the central nervous and immune systems. This minireview updates the most important evidence on the role of CMA in tissue homeostasis and disease pathogenesis, highlighting CMA as an essential mechanism to understand for the development of future treatments in neurodegenerative diseases, cancer, and other pathologies.

## 2. Chaperone-Mediated Autophagy

Autophagy embraces the lysosomal degradation and recycling of a diverse range of intracellular components, including proteins, organelles, and even invading pathogens. Autophagy plays an essential role in maintaining proteostasis and organelle turnover, but also has a pivotal role in regulating cellular energetics, stress responses, pathogen defense, cellular reprogramming, stem cell maintenance, and other fundamental cellular mechanisms.

In most mammalian cells, three major forms of autophagy coexist, namely macroautophagy, microautophagy, and CMA [1]. The distinctive features of CMA include its specificity toward substrates, which are exclusively proteins. Unlike macroautophagy, CMA does not rely on autophagosome formation; instead, the substrates are directly translocated into the lysosomes for degradation through a lysosomal–membrane complex.

### 2.1. Mechanisms of CMA

CMA’s unique features primarily stem from its substrate recognition and targeting mechanisms. The KFERQ-like motif present in the substrate protein is necessary and sufficient for targeting the substrate to CMA degradation [2]. The canonical CMA motif consists of the pentapeptide KFERQ, but alternative, well-defined modifications to this pentapeptide, with specific alternative amino acids, can also generate alternative KFERQ-like motifs [2,3,4,5,6,7,8]. Approximately 40% of proteins in the mammalian proteome contain a recognized KFERQ-like motif. Additionally, post-translational modifications, such as phosphorylation or acetylation, can generate or complete the KFERQ-like motif, increasing the number of potential CMA substrates. Additionally, post-translational modifications outside the motif can also modulate CMA targeting by facilitating conformational changes in the protein that expose or mask the motif and trigger or avoid CMA-mediated degradation of the protein.

Hsc70 (heat-shock cognate protein 70) is the chaperone responsible for recognizing and binding the KFERQ-like motif present in the substrate protein. With the help of additional co-chaperones, including HSP90, HSP40, HOP, HIP, and BAG-1, the complex substrate–Hsc70 is targeted to the lysosomal membrane to interact with the transmembrane protein LAMP-2A (Lysosome-associated membrane protein type 2A) [9,10] (Figure 1).

Lysosomal LAMP-2A stands as the pivotal player in CMA. There are three LAMP2 splicing variants (LAMP-2A, LAMP-2B, and LAMP-2C), differing only in the small cytosolic tail exposed outside the lysosome [11]. Only the LAMP-2A variant is essential for CMA since its cytosolic tail is required for the lysosomal docking of the Hsc70–substrate complex. Under resting conditions, LAMP-2A is found as a monomer and is mainly located in lipid-enriched microdomains within the lysosomal membrane [12]. The interaction of the Hsc70–substrate complex facilitates the assembly of the CMA translocation complex, where LAMP-2A proteins join in homotrimeric structures. This assembly creates a channel through which the substrate protein unfolds and is translocated into the lysosomal lumen, initiating its degradation (Figure 1).

The LAMP-2A abundance on the lysosomal membrane is the limiting factor for CMA and is primarily regulated by the dynamic distribution and turnover of the LAMP-2A protein rather than de novo synthesis. LAMP-2A monomers can be cleaved and degraded when monomeric LAMP-2A is inside lipid microdomains, subsequently downregulating CMA. Conversely, the exclusion of LAMP-2A from lipid microdomains prevents cleavage and facilitates the assembly of LAMP-2A multimers necessary for CMA substrate uptake and an increase in CMA activity. Changes in the lipid composition of the lysosomal membrane can affect the stability of the translocation complex and the levels of LAMP-2A [13,14,15].

In addition to the modulation of LAMP-2A levels in the lysosomal membrane, CMA activity is tightly controlled through other signaling pathways that can upregulate or downregulate CMA in response to changing cellular conditions.

Most of these mechanisms of CMA regulation are based on the transcriptional modulation of LAMP-2A (Figure 1). Unlike its role in macroautophagy, retinoic acid receptor alpha (RARα) signaling negatively regulates the CMA pathway. RARα signaling inhibits CMA activity by decreasing the expression of LAMP-2A, Rab11, and Rab-interacting lysosomal protein (RILP) [16]. Subsequently, genetic or pharmacological inhibition of RARα can promote LAMP-2A trafficking to lysosomes and promote CMA activity. Recently, different inhibitors and antagonists of RARα compounds like AR7, GR2, QX77, CA77.1, and CA 39, have been shown to be good pharmacological CMA activators in different models of neurodegenerative diseases, both in vitro and in vivo [17,18,19,20].

TFEB is a master regulator of lysosomal biogenesis and autophagy. It controls the expression of many autophagy-related genes, although previous studies demonstrated that LAMP-2A is not transcriptionally regulated by TFEB [21]. Recent works suggested that TFEB activation could enhance LAMP-2A levels and CMA activity, as well as the inhibition of TFEB translocation to the nucleus through phosphorylation, which can downregulate LAMP-2A transcription and CMA activity [22,23].

### 2.2. Selective Regulation in Response to Stress and Cellular Conditions

Under specific conditions or stressors like oxidative stress, genotoxic damage, or hypoxia, LAMP-2A levels can be selectively upregulated in response to stress and cellular conditions [24,25,26,27]. For instance, the generation of reactive oxygen species (ROS) during T cell activation promotes the nuclear translocation of the nuclear factor of activated T cells-1 (NFAT1) that directly binds the *Lamp2* promoter region, increasing *Lamp2a* expression [24] and, consequently, increasing CMA. Furthermore, this CMA activation triggers the degradation of two signaling inhibitors of T cell activation, Rcan-1 and Itch, (Y85,87,117), endorsing the process of T cell activation, proliferation, and differentiation.

Under oxidative stress, the Nrf2–Keap1–ARE signaling mechanism is activated as an antioxidant response [28] and Nrf2 promotes the transcription and expression of LAMP-2A and activates CMA. This promotion of *Lamp2a* expression and CMA activity can be achieved with the pharmacological activation of Nrf2 and downregulated when Nrf2 knockdown is performed [29].

Beyond the transcriptional regulation of LAMP-2A, the mTORC2–AKT1–PHLPP1 axis exerts the dual regulation of CMA by acting on AKT1 [30]. This axis stands out for its distinctive localization on the lysosomal membrane compared to other pathways that rely on cytoplasmic signals. Moreover, its signaling depends on phosphorylation/dephosphorylation mechanisms rather than the transcriptional regulation of LAMP-2A. mTORC2, localized into the lysosomal membrane, phosphorylates downstream AKT1, promoting GFAP phosphorylation. This process stabilizes GFAP, preventing its binding to the LAMP-2A– Hsc70 complex and leading to an inhibitory effect on CMA [30,31]. In contrast, PHLPP1 dephosphorylates AKT, which promotes GFAP dephosphorylation and enhances the assembly and disassembly of LAMP-2A, thus boosting CMA activity [30] (Figure 1).

## 3. Biological Functions of CMA

### 3.1. Protein Quality Control

CMA basal activity, as a protein quality control mechanism in most cells and tissues, contributes to proteome remodeling through its circadian properties, degrading misfolded or damaged proteins to maintain their localization, levels, and conformation [25,32,33,34,35]. Although in some cases the failure of CMA due to aging or the persistence of stress stimuli can be solved with bi-directional compensatory mechanisms such as macroautophagy or the ubiquitin–proteasome system (UPS) [36], in neurons, activated T cells, and hematopoietic stem cells [37,38], there is not any compensatory crosstalk among mechanisms. In these cases, damaged proteins are accumulated, leading to cellular stress due to proteotoxicity [20,39,40]. Furthermore, uncompensated CMA failure also drives unfunctional proteins due to protein aggregation because of protein accumulation [41]. That is the case of pathogenic proteins related to protein conformational disorders [42], such as LRRK2, alpha α-synuclein, and tau, which are aggregated at the lysosomal surface by CMA failure, and subsequently exacerbate the effects of the CMA impairment, promoting a proteotoxicity proteome [22] (Figure 2).

### 3.2. Cellular Energetics

CMA also has regulatory functions, degrading functional proteins to finish their function and regulate different cellular processes. In fact, CMA has a role in the maintenance of cellular energetics, degrading enzymes implicated in glucose hydrolysis, in addition to inactive forms, to maintain glucose levels during starvation [8,43]. Furthermore, CMA regulates lipid metabolism, degrading lipogenic enzymes, regulator proteins of lipid mobilization, and limiting enzymes in fat acid metabolism to prevent the lipid accumulation [41,43,44,45,46]. Additionally, CMA also controls lipid droplet mobilization, eliminating enzymes that regulate lipogenesis through lipid uptake/synthesis [45,46], and ensuring efficient lipolysis, such as in the case of PLIN2 and PLIN3 [47].

On the other hand, CMA prevents mitochondrial dysfunction and regulates dynamics through protein and enzyme degradation, including Krebs enzymes and proteins contributing to mitochondrial metabolism in the cells [43,48] (Figure 2).

### 3.3. Cellular Reprogramming

CMA has an important role in cellular reprogramming through its selective regulatory function of remodeling the proteome. It participates in the metabolic switch needed for hematopoietic stem cell (HSC) activation, and its failure in old organisms impairs hematopoietic function and HSC Proliferation [41]. Furthermore, it controls the self-renewal/differentiation of embryonic stem cells (ESCs), modulating epigenome changes through the degradation of enzymes implicated in the α-ketoglutarate metabolism [49]. Activated CMA degrades isocitrate dehydrogenase (IDH) 1 and IDH2, leading to the reduction in the intracellular levels of α-ketoglutarate, which are involved in the pluripotency maintenance of mouse ESCs. Thus, in quiescent ESC, as CMA is inactive through the *Lamp2a* gene silencing through binding of the transcriptional factors OCT4 and SOX2 to a distal promoter region, IDH1 and IDH2 are accumulated and promote high levels of α-ketoglutarate. During cell differentiation, the inactivation of CMA and subsequent metabolic changes that determine the self-renewal of ESCs are prevented through the silencing of SOX2 and OCT4 genes [49].

Additionally, CMA activity regulates cell differentiation in different systems, and its failure with age leads to consequent cell alterations that can enhance significant inflammatory lesions in the organism [46,50]. The blockage of CMA in vascular smooth muscle cells leads to transdifferentiation [46], whereas in rat mesenchymal stem cells, it induces osteoblast differentiation [50].

CMA also regulates the adaptive immune response through the recycling of negative regulators of T cell activation such as ITCH and RCAN. Thus, its deficiency in aged T cells impairs T cell function against pathogens such as bacterial infection [24].

On the opposite, an aberrant increase in CMA flux in B cells seems to be responsible for the abnormal B cell responses in lupus autoimmune diseases [51] (Figure 2).

### 3.4. Other Cellular Processes

CMA also has a role in cellular remodeling through its selective function of regulating cell growth and survival through the degradation of the transcription factors PAX2, MEF2D, MEF2A, and the HMBG1 [52]. Furthermore, CMA activity contributes to regulating the cell cycle in response to stress, timely degrading cell cycle modulators such as Chk1, RND3, HIF-1α, and TP73 [53,54]. In fact, CMA also participates in circadian rhythm regulation, eliminating the transcription factors BMAL1, CLOCK, and REVERBα [32] (Figure 2).

### 3.5. Cellular Defense

It is important to highlight another important function of CMA as a cellular defense. CMA regulates cell death by contributing to the prevention of stress-induced apoptosis [39,55], but in contrast, promoting ferroptosis [56]. ER stress-induced CMA is essential for maintaining cellular homeostasis and protecting cells from cell death through the regulation of the unfolded protein response [39,40]. Furthermore, CMA activity in response to other stressors suppresses apoptosis, mediating the degradation of damaged proteins [55], however, it is required to enhance ferroptosis in some contexts through the degradation of the glutathione peroxidase (GPX4) [56].

Moreover, the regulation function of the mitochondrial dynamics and function by CMA may also regulate cell death [48]. CMA regulates mitochondrial dynamics through the degradation of MARCHF5, a ubiquitin ligase required for mitochondrial fission. Thus, CMA prevents mitochondrial dysfunction due to the excessive fragmentation of mitochondria [48].

Indeed, CMA has a defense function in macrophages, degrading the enzyme TRIM21 to prevent cell death in response to pathogens such as bacterial infection [57]. In addition, CMA modulates the innate immune response, reducing antiviral immune responses and enhancing viral infection through the degradation of key inflammation factors, such as NLRP3 [45], TBK1 [58], STING [59], and other specific regulator proteins [59,60] (Figure 2).

## 4. CMA and Human Diseases: Role in Pathogenesis

### 4.1. Neurodegenerative Diseases

A mounting body of evidence underscores the crucial involvement of proteolytic pathway dysregulation in the initiation and/or progression of neurodegenerative diseases [53]. Certain unfolded, modified, damaged, or mutant proteins exhibit a tendency to aggregate, and considering that neurons are post-mitotic cells, they rely on intracellular degradative mechanisms to handle protein accumulation and aggregation. The failure of these cleaning systems and the subsequent accumulation of toxic aggregates represent a common hallmark in numerous neurodegenerative disorders.

During aging, a decrease in CMA activity caused by a decrease in LAMP-2A levels has been found in mammals in almost all cell types and tissues [24,36,41,61,62]. Recently, the contribution of CMA to the maintenance of the metastable neuronal proteome, under physiological conditions, highlighted the essential role of CMA in neuronal homeostasis [20] and showed that in the central nervous system, both macroautophagy and CMA display non-redundant functions. Indeed, CMA is instrumental in avoiding neuronal proteotoxicity and guaranteeing proteostasis maintenance.

Consequently, CMA failure due to aging or due to the presence of pathogenic proteins associated with neurodegenerative diseases has a strong impact on the neuronal proteome, promoting protein insolubility and aggregate formation.

#### 4.1.1. Parkinson’s Disease (PD)

PD is characterized by dopaminergic neuron loss and abnormal protein aggregate accumulation, known as Lewy bodies (LB), where the primary component is aggregated α-synuclein [63]. Among neurodegenerative diseases, PD was the first associated with CMA because α-synuclein was one of the first substrates identified as a CMA substrate and, although alternative proteolytic pathways can also contribute to α-synuclein degradation, the neuron primarily relies on the CMA pathway for its physiological turnover. Alterations in CMA activity directly impact α-synuclein levels, with the loss of CMA activity linked to α-synuclein accumulation in numerous in vitro and in vivo PD models, as well as in samples derived from PD patients [64,65,66].

The observed failure of CMA activity in PD may stem from multiple causes, including the age-dependent decline of CMA and the presence of pathogenic proteins associated with the disease. For instance, mutant variants of α-synuclein, such as A53T and A30P, interact with LAMP-2A, hindering the CMA translocation complex on the lysosomal membrane, impeding the internalization of synuclein, and compromising the CMA degradation of other substrates [67]. This situation exacerbates the accumulation of toxic aggregated proteins, contributing to neurodegeneration. Post-translational modifications of α-synuclein, like phosphorylation and dopamine modification, also interfere with CMA, preventing their own degradation and the CMA-dependent degradation of various protein substrates [68].

CMA defects extend beyond α-synuclein; other PD-related proteins, including UCH-L1 and LRRK2, can also impact CMA function. UCH-L1 and LRRK2 pathogenic variants directly interact with the LAMP-2A translocation complex blocking CMA in an analogous mechanism as mutant synuclein. Other PD-related proteins can affect CMA activity without directly interacting with LAMP-2A. For example, Vacuolar protein sorting-35 (VPS35) is essential for the endosome-to-Golgi retrieval of LAMP-2A; mutations or deficiency can promote LAMP-2A degradation in dopamine neurons, contributing to PD pathogenesis [69].

The first genetic risk factor for PD is the presence of a mutation in the GBA gene [70,71]. GBA encodes β-glucocerebrosidase (β-GCase or GBA), a lysosomal enzyme with the primary function of hydrolyzing GlcCer into ceramide and glucose. Mutations associated with PD result in a loss of GCase activity and the consequent accumulation of GBA substrates, including sphingolipids and other lipids like gangliosides and cholesterol, all of them the primary components of lipid microdomains in the lysosomal membrane. These changes in lysosomal lipid components favor the presence of LAMP-2A as a monomer, preventing the assembly of the CMA–translocation complex, inhibiting CMA activity, and promoting α-synuclein accumulation [15]. Restoring CMA function has been explored as a potential therapeutic target for PD. Increasing CMA activity or upregulating key CMA components such as LAMP-2A has been shown to mitigate neurotoxicity associated with α-synuclein and protect against dopaminergic neuron loss: genetic overexpression of *Lamp2a* has demonstrated clearance of α-synuclein in dopaminergic neurons and reduced cell loss in vivo [72] and, pharmacologically, the use of alphaRAR inhibitors like AR7 and QX77 or other CMA activators like the peptide Humanin has been shown to enhance LAMP-2A levels and CMA activity in various in vitro PD models [15,18,19].

#### 4.1.2. Alzheimer’s Disease (AD)

AD is characterized by the accumulation of Aβ plaques and intracellular tau tangles. CMA activity has been shown to decrease in Alzheimer’s disease, and several AD-associated proteins have been shown to interact with CMA.

CMA is involved in the degradation of wild-type tau, but mutant forms of tau can bind to LAMP-2A and disrupt its lysosomal membrane translocation, impairing CMA activity [20].

Additionally, other AD-related proteins, such as RCAN1, are degraded by CMA as part of the regulatory mechanism to prevent excessive tau phosphorylation. The loss of CMA can indirectly promote the hyperphosphorylation of tau [73,74].

Like in other neurodegenerative diseases, the modulation of CMA has been shown to have therapeutic benefits in AD management. The pharmacological CMA activator CA77.1 (an AR7 analog and based on the inhibition of RARα) has been shown to present beneficial effects in AD-related pathology in two different in vivo AD mouse models [20].

Amyloid-beta precursor protein (APP) has also been identified as a substrate for CMA, revealing its interaction with Hsc70 in a manner dependent on IKKα/β. The authors proposed metformin as a new CMA activator treatment, via activation of TAK1-IKKα/β signaling that leads to the phosphorylation of Hsc70. Activating CMA through Hsc70 overexpression or the administration of metformin substantially diminished the accumulated brain Aβ plaque levels in an APP/PS1 mouse model [75].

#### 4.1.3. Other Neurodegenerative Diseases

Huntington’s disease (HD) and other neurodegenerative disorders, such as prion diseases and frontotemporal dementia (FTD) are also related to CMA dysfunction through disease-related proteins that have been confirmed as CMA substrates. In HD, mutant huntingtin protein (Htt) is identified as a CMA substrate and interacts with CMA machinery. Additionally, CMA activity is initially upregulated in the early stages of HD as a compensatory response to macroautophagy impairment, however, it decreases with age, contributing to cellular failure and the onset of pathological manifestations [76,77,78]. CMA is also implicated in the degradation of PrP in prion diseases and TDP-43 in ALS and FTD. Mutations or dysfunctions in various proteins, including PLK3 and SCA21-related TMEM240, affect CMA activity in these disorders [79,80].

We cannot discard that numerous proteins linked to neurodegeneration might be CMA substrates. In many of these circumstances, the failed CMA does not solely impact the individual degradation of a specific disease-associated protein, but also adds to the overall failure of CMA, thereby contributing to neuronal proteotoxicity.

The involvement of CMA dysfunction in various neurodegenerative diseases has been substantiated by the therapeutic impact of activating CMA to counteract the pathogenic phenotype. Recently, the validation of pharmacological CMA activation, achieved through AR7 analogs termed CMA activators (CAs), was observed in vivo across diverse neurodegenerative models including Alzheimer’s disease (AD), as previously discussed [75], and in a mouse model of retinitis pigmentosa [17]. In this model, characterized by the progressive degeneration of neurons in the retina, treatment with CA compounds (CA39 and CA77) effectively alleviated retinal degeneration, underscoring their potential as a therapeutic strategy against neuronal degeneration in vivo and validating their promise for addressing other neurodegenerative diseases.

### 4.2. Cancer

#### 4.2.1. CMA Regulation: Pro-Tumor versus Anti-Tumor Roles

As CMA activity is needed for different cellular functions in the cells, its failure with age or any dysregulation of its physiological activity levels leads to different affected physiological processes causing diseases such as cancer [24,41,45,46,81]. During aging, LAMP-2A levels at the lysosomal membrane are less stable, which promotes deficient CMA activity, consequently leading to deficient control of the protein quality in the cells and, therefore, malignant cell transformation [82,83]. Importantly, CMA in basal levels is needed to maintain cell homeostasis and prevent malignant cell transformation by increasing proteostasis and avoiding DNA damage [84]. However, if CMA activity is dysregulated, has a pro-oncogenic role, and transformed cells aberrantly upregulate CMA activity, eliminating several anti-tumor proteins that negatively regulate tumor cell survival and proliferation [82,85,86]. Moreover, CMA has a role in the regulation of cellular energetics with a pro-oncogenic function, modulating metabolic switches that contribute to tumor growth and survival [82]. CMA constitutively upregulated in different types of cancer cells is needed to degrade glycolytic enzymes required to induce the anaerobic glycolysis which is essential to support the Warburg metabolism in cancer cells for tumor progression [82,85]. However, functional CMA is needed to sustain the Warburg effect in cancer cells. As has been demonstrated in melanoma cells and lung cancer, CMA blockage stabilizes P53 which, in contrast, decreases the transcriptional levels of glycolytic enzymes [85]. Additionally, increased CMA sustains cancer cells, affecting many cellular processes that control protein synthesis through the degradation of translation components [87].

#### 4.2.2. Identified Substrates in Different Types of Cancers

In recent years, more and more different protein substrates have been identified as CMA substrates to prevent the transformation of healthy cells through their anti-oncogenic function. That is the case of the proto-oncogenic protein mouse double-minute 2 homologue (MDM2) [88] and the translational-controlled tumor-associated protein (TCTP) [89], whose reduced levels in the cell through CMA activity seem to be related to the prevention of the spontaneous liver tumors that appear with the age [36]. CMA regulates the transcription factors that regulate cell proliferation and apoptosis such as the paired-box protein PAX2 [90]. And it is also involved in the regulation of the transcription factor c-Myc, since it degrades the protein phosphatase 2A (CIP-2A), which is responsible for the degradation of this proto-oncogene [84]. Moreover, the P65, a component of the nuclear factor-κB, is degraded by CMA activity preventing the increase in NF-κB signaling in epithelial cells and a subsequent epithelial–mesenchymal transition that leads to tumorigenesis [91]. Hexokinase II and pyruvate kinase-II, glycolytic enzymes required for tumorigenesis, are regulated by degradation through CMA in some types of cancers [8,92], but specific post-translational modifications in those proteins affect their degradation, leading to pro-oncogenic effects that support CMA function in the energy maintenance of cancer cells. CMA is involved in DNA repair mechanisms and the re-entry of cells into the cell cycle. In response to DNA damage, CMA degrades activated Chk1, preventing a persistent activation and induction of genome instability as it occurs in cancer cells [6]. Conversely, CMA has been shown to be essential in regulating cell proliferation and promoting tumor cell invasion in gastric cancer, degrading RND3, a Rho family GTPase, an anti-proliferative protein related to preventing tumorigenesis and metastasis [7]. In addition, the nerve growth factor (NGFR), which is upregulated in several types of cancers, promotes the survival of transformed cells, binding to the tumor suppressor p53, which enhances its degradation by CMA [93]. Moreover, the hypoxia-inducible factor1 (HIF-1), a CMA substrate, is another transcription factor that controls gene transcription under hypoxic conditions, leading to cell proliferation in healthy cells, whereas if it is degraded by cyclin-dependent kinase 2-induced CMA activity in some types of cancer cells, tumor cell proliferation in enhanced [26].

#### 4.2.3. Targeting CMA in the Tumor Microenvironment

CMA has been identified in different stages of tumors and is highly induced for tumor cell proliferation and survival in multiple types of cancers, including renal, prostate, breast, lung, colon, melanoma, and hepatocellular and cervical carcinoma [82,85,94,95]. LAMP-2A has been also suggested as a potential prognosis marker in glioblastoma [96], gastric cancer [7] and renal carcinoma [95], as its expression is correlated with the high CMA activity needed to maintain the metabolic requirements of cancer stem cells which initiate tumorigenesis.

Importantly, the tumor cell not only just upregulates *Lamp2a* expression and, therefore, increases CMA activity for its proliferation and survival, but it even promotes aberrant upregulation in the cells of the tumor microenvironment for its own benefit [86,97]. That is the case of perivascular cells in glioblastoma cancer, known as pericytes [97], which are also located in other microvascularized cancers and show an immunosuppressive pro-tumor function [98]. Stable tumor cell–pericytes interactions, and the ablation of the anti-tumor immune function of pericytes driven by aberrantly increased CMA activity, promote tumor cell proliferation and survival [99,100].

All these findings lead us to think that CMA could be a good target for cancer. However, the role of CMA in certain stages of the tumor and the need for it to be functional in certain cells to prevent their transformation [94], including its essential role in different cell types of the cellular microenvironment useful to eliminate the tumor [99], makes it critical to better understand the physiological and pathological functions of CMA in different types of cells. It is necessary to consider the type of tumor and stage to develop selective molecules against CMA activity. Currently, nothing is known other than some CMA-modulating molecules that make it deficient without eliminating it completely, and that may have undesirable effects since they can also affect other mechanisms of the cell, without being selective [82,86,101].

### 4.3. Aging-Associated Diseases and Other Pathologies

CMA activity presents a protective role in bronchial epithelial cells through increased *Lamp2a* expression dependent on activated Nrf2 and in response to oxidative stress such as cigarette smoke [40]. Chronic obstructive pulmonary disease (COPD) is a representative, aging-associated pulmonary disease, based on persistent airflow limitation with an abnormal inflammatory response related to harmful gases and particles [102]. The deficient CMA activity due to reduced levels of LAMP-2A, as occurs in COPD, negatively affects the bronchial epithelial cell survival and, subsequently, pulmonary function [40,102].

Aging-associated metabolic disorders, such as liver diseases, have also been reported to be related to affected CMA activity [36,43,103,104]. CMA protects hepatocytes from lipotoxicity and oxidative stress in normal conditions [43,44]. In contrast, reduced levels of LAMP-2A and other positive regulators of CMA in the liver of non-alcoholic and alcoholic fatty liver patients, including hepatic steatosis, drive deficient CMA that unbalances the lipid metabolism in response to oxidative stress [103,104,105].

CMA failure can also be implicated in cardiovascular diseases, as reduced LAMP-2A levels in vascular smooth muscle cells and macrophages promote risks and severity of atherosclerotic plaques and exacerbated pro-inflammatory function. In addition, aging can aggravate these effects due to the deficient CMA activity that deregulates lipid metabolism and impairs NLPR3 inflammasome degradation [45,46].

On the other hand, dysfunction of CMA as a consequence of some diseases can aggravate the pathology. That is the case in the involvement of suppressed CMA activity in several pathologic conditions in the kidney [106]. The accumulation of CMA substrates by impaired CMA in the renal cortex during acute diabetes mellitus might also be associated with the development of diabetic-induced renal hypertrophy [106,107].

CMA dysfunction is also implicated in the pathogenesis of Mucolipidosis type IV (MLIV), a lysosomal storage disorder, as its activity is deficient in fibroblasts from these patients [108].

CMA also plays an important role in immune responses against pathogens [24,57], and its dysfunction by *Salmonella Typhimurium*, results in impaired protection against oxidative stress in the infected macrophages that subsequently leads to cell death as a key virulence strategy [57]. Moreover, CMA failure in T cells sensitizes them to pathogen infection, such as Listeria and thus, with age, the T cell effector response against bacterial infection is more vulnerable [24]. In contrast, CMA is aberrantly upregulated in B cells during the autoimmune disease lupus, enhancing the inflammatory effects that contribute to pathogenesis [51].

CMA is also involved in the pathogenesis of ulcerative colitis, through its aberrant induction led by increased protein expression levels of LAMP-2A, which subsequently leads to inflammatory symptoms of bowel disease [109]. In contrast, induced CMA activity prevents the pathogenesis of intestinal fibrosis through inhibition of the epithelial-mesenchymal transition through the downregulation of NF-κB (p65/RelA) signaling [91].

## 5. Conclusions

More and more findings are emerging with CMA as an attractive therapeutic target to treat multiple pathologies, attempting to restore its activity levels by modulating its components or restoring normal levels of the lysosomal receptor LAMP-2A, which is affected by aging. However, in many other inflammatory pathologies, including cancer, CMA levels are abnormally overactivated, and what would be of interest is to suppress the expression of LAMP-2A or modulate the activity of CMA through other components, correcting it to physiological levels.

Although the regulators of CMA to modulate the pathway are increasingly better known, more studies are needed to deepen knowledge of the dynamics of substrate translocation and new physiological functions attributed to CMA. A better understanding of CMA regulatory roles in different systems and tissues, including the consequences of their loss with aging and pathophysiology, would facilitate possible future treatments for multiple diseases.

## Figures and Tables

**Figure 1 biomedicines-12-00257-f001:**
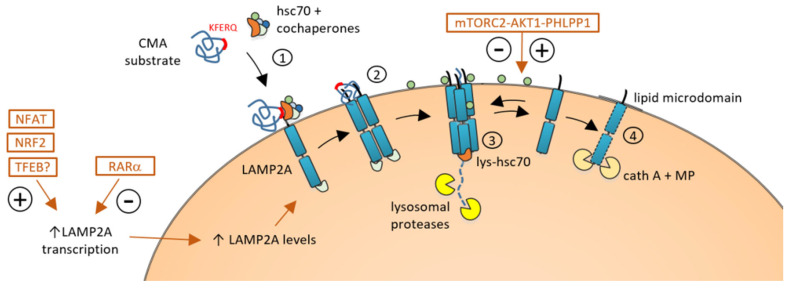
Chaperone-mediated autophagy (CMA) mechanism and regulation: CMA substrates carrying the KFERQ-like motif are recognized by chaperone Hsc70 and co-chaperones (1) and delivered to the lysosomes where the complex interacts with LAMP-2A, promoting the assembly of the translocation complex (2). Assisted by lysosomal Hsc70 (lys-Hsc70), the substrate is unfolded and translocated inside the lysosomal lumen, where it is degraded by lysosomal proteases (3). Dissociation of LAMP-2A from the translocation complex favors its turnover that occurs in lipid microdomains by the action of cathepsin A and a metalloproteinase (MP) (4). CMA is regulated by different cytosolic signals that can promote (+) or inhibit (−) LAMP-2A transcription (left) and by the mTORC2-AKT1-PHLPP1 axis on the lysosomal membrane that both promote (+) and inhibit (−) the assembly or disassembly of the CMA translocation complex.

**Figure 2 biomedicines-12-00257-f002:**
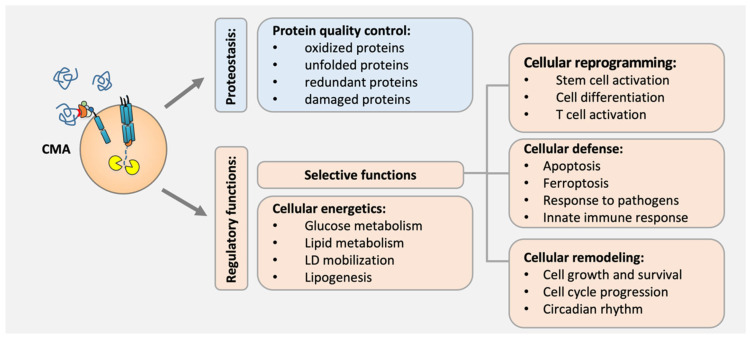
Biological functions of CMA and its physiological role in cells. CMA participates in the protein quality control of cells through the degradation of nonfunctional, damaged, or unfolded proteins. CMA also has regulatory functions contributing to cellular energetics, among others, such as the regulation of cellular reprogramming, defense, and remodeling, which selectively modulates the proteome to regulate several processes.

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
