# Peer review of "The Role of Chaperone-Mediated Autophagy in Tissue Homeostasis and Disease Pathogenesis"

_biomedicines, 2024, doi:10.3390/biomedicines12020257_

Round 1

Reviewer 1 Report

Comments and Suggestions for Authors

The authors overviewed molecular mechanisms of Chaperone-Mediated Autophagy (CMA) and its contribution to major neurodegenerative diseases such as Parkinson’s disease, Alzheimer’s disease, and cancers. This article also highlighted the therapeutic potential of CMA-targeted treatment for the diseases mentioned above. The research area of CMA is one of the mature but still rigorously expanding fields. This article will attract a wide range of audiences including cell biologists and basic biomedical researchers interested in pathogenetic mechanisms at the molecular level. From an objective viewpoint, this review would be informative for experts and serve as the foundation for researchers entering this research field. The authors can improve this article by addressing the following issues. Hopefully, these comments help craft a better-quality paper.

  1. The abstract and Introduction are identical in the current version of this manuscript. I would guess that the special issue is going to begin with this article because the authors are guest editors. I would recommend that Abstract and Introduction will be edited according to the contents of the other papers published in this special issue. One of the paper recapitulate the crucial points of the entire special issue would help readers obtain further insights into this topic. I would suggest editing this review article to represent all of this issue, particularly, if the abstract part serves as a concise review of the entire issue would be great.

  1. I would recommend editing the title, “Biological Functions of CMA: Physiological Role”. The word “Physiological” conveys the impression of an in vivo study. I interpreted this part to explain the crosstalk between CMA and the other cell biological processes. It would be great if the title could direct readers’ attention to the cellular processes. “Physiological” can be misleading.

  1. L169-173, It would be better if you could state more details. You might mention specific molecular targets of CMA during stem cell self-renewal and differentiation. And readers would like to understand the molecular mechanisms of how CMA can modulate epigenomics through alpha-ketoglutarate metabolism.

  1. “Cellular Remodeling” might be vague and not capture the contents of this part. I would recommend finding the other words or “Other cellular processes” could also be fine.

  1. L190-191, readers might want to know the molecular details of this part as well. How does the CMA regulate apoptosis, ferroptosis, and mitochondrial dynamics?

  1. In the 4.3 Aging-Associated and Other Pathologies, it seems that two logics are mixed up here. This part describes two topics. One is CMA deficiency causes some diseases, and another is dysfunction of CMA is the consequence of some diseases. These two different logics should be well-explained in an organized fashion. I would recommend taking a paragraph for each topic and separately describe.

  1. I would strongly recommend adding another chapter focussing on how aging influences the CMA. This chapter might serve very well to understand the causal effects of CMA on age-related diseases such as PD and AD since the relevance of CMA on the sporadic pathogenesis of these diseases is not discussed.

  1. Figure 1. It would be hard for me to distinguish positive and negative regulations, particularly cartoons of mTORC2-AKT1-PHLPP1. And I recommend adding descriptions of all molecules in a dedicated area.

  2.  
  1. Figure 2. I would recommend adding the major target proteins of CMA in each cellular process and their citations. It would be better if you organize these data within a table. My suggestion is that this information is not necessarily listed in a figure.

Author Response

Thank you for considering the revision of our manuscript. We have addressed each concern raised by the reviewers as listed below:

REVIEWER 1:

Comments and Suggestions for Authors

The authors overviewed molecular mechanisms of Chaperone-Mediated Autophagy (CMA) and its contribution to major neurodegenerative diseases such as Parkinson’s disease, Alzheimer’s disease, and cancers. This article also highlighted the therapeutic potential of CMA-targeted treatment for the diseases mentioned above. The research area of CMA is one of the mature but still rigorously expanding fields. This article will attract a wide range of audiences including cell biologists and basic biomedical researchers interested in pathogenetic mechanisms at the molecular level. From an objective viewpoint, this review would be informative for experts and serve as the foundation for researchers entering this research field. The authors can improve this article by addressing the following issues. Hopefully, these comments help craft a better-quality paper.

  1. The abstract and Introduction are identical in the current version of this manuscript. I would guess that the special issue is going to begin with this article because the authors are guest editors. I would recommend that Abstract and Introduction will be edited according to the contents of the other papers published in this special issue. One of the paper recapitulate the crucial points of the entire special issue would help readers obtain further insights into this topic. I would suggest editing this review article to represent all of this issue, particularly, if the abstract part serves as a concise review of the entire issue would be great.

We appreciate the reviewer's observation and constructive suggestion regarding the abstract and introduction sections.

In response, we have made revisions to both the abstract and introduction, emphasizing that this manuscript serves as a "minireview" specifically focused on the topic covered by the Special Issue.

  1. I would recommend editing the title, “Biological Functions of CMA: Physiological Role”. The word “Physiological” conveys the impression of an in vivo study. I interpreted this part to explain the crosstalk between CMA and the other cell biological processes. It would be great if the title could direct readers’ attention to the cellular processes. “Physiological” can be misleading.

We understand and appreciated the suggestion, we have changed the title to “Biological Functions of CMA”. 

  1. L169-173, It would be better if you could state more details. You might mention specific molecular targets of CMA during stem cell self-renewal and differentiation. And readers would like to understand the molecular mechanisms of how CMA can modulate epigenomics through alpha-ketoglutarate metabolism.

Text: Furthermore, it controls self-renewal/differentiation of embryonic stem cells, modulating epigenome changes by degradation of enzymes implicated in the -ketoglutarate metabolism. Additionally, CMA activity regulates cell differentiation in different systems, and its failure with age, leads to consequent cell alterations that can enhance significant inflammatory lesions in the organism [50,53].

We value the insightful comments provided by the reviewers. In response, we have revised this section and incorporated the additional information as suggested:

¨Activated CMA degrades Isocitrate dehydrogenases (IDH) 1 and IDH2 leading to the reduction of the intracellular levels of -ketoglutarate, which are involved in the plu-ripotency maintenance of mouse ESCs. Thus, in quiescent ESC, as CMA is inactive through the Lamp2a gene silencing by binding of the transcriptional factors OCT4 and SOX2 to a distal promoter region, IDH1 and IDH2 are accumulated and promote high levels of -ketoglutarate.  During cell differentiation, inactivation of CMA and subse-quent metabolic changes that determine self-renewal of ESCs, are prevented through silencing of SOX2 and OCT4 genes [53].

Additionally, CMA activity regulates cell differentiation in different systems, and its failure with age, leads to consequent cell alterations that can enhance significant in-flammatory lesions in the organism [50,54]. Blockage of CMA in vascular smooth muscle cells lead to transdifferentiation [50], whereas in rat mesenchymal stem cells induces osteoblast differentiation [54]¨.

  1. “Cellular Remodeling” might be vague and not capture the contents of this part. I would recommend finding the other words or “Other cellular processes” could also be fine.

We have changed the title of this section according to the suggestion.

  1. L190-191, readers might want to know the molecular details of this part as well. How does the CMA regulate apoptosis, ferroptosis, and mitochondrial dynamics?

Text: Moreover, the regulation function of the mitochondrial dynamics and function by CMA, may also regulate cell death [52].

We value the insightful comments provided by the reviewers. In response, we have revised this section and incorporated the additional information as requested:

¨ER stress-induced CMA is essential for maintaining cellular homeostasis and protecting cells from cell death through the regulation of the unfolded protein response[41,42]. Furthermore, CMA activity in response to other stressors suppresses apoptosis mediating the degradation of damaged proteins [60], however, is required to enhance ferroptosis in some contexts through degradation of the glutathione peroxidase (GPX4) [61].  

Moreover, the regulation function of the mitochondrial dynamics and function by CMA, may also regulate cell death [52]. CMA regulates mitochondrial dynamics by the degradation of MARCHF5, an ubiquitin ligase required for mitochondrial fission. Thus, CMA prevents mitochondrial dysfunction due to the excessive fragmentation of mitochondria [52]¨.

  1. In the 4.3 Aging-Associated and Other Pathologies, it seems that two logics are mixed up here. This part describes two topics. One is CMA deficiency causes some diseases, and another is dysfunction of CMA is the consequence of some diseases. These two different logics should be well-explained in an organized fashion. I would recommend taking a paragraph for each topic and separately describe.

We appreciate the comment and agree with the suggestion, accordingly we have reorganized this section and emphasized this observation.

  1. I would strongly recommend adding another chapter focussing on how aging influences the CMA. This chapter might serve very well to understand the causal effects of CMA on age-related diseases such as PD and AD since the relevance of CMA on the sporadic pathogenesis of these diseases is not discussed.

The decline of CMA during aging and the relevance on age-related diseases was briefly commented in section 4.1. However, to further emphasize this critical aspect, we have added a new sentence and additional references. We believe these modifications will strengthen the connection between CMA, aging, and its potential implications for age-related diseases.

  1. Figure 1. It would be hard for me to distinguish positive and negative regulations, particularly cartoons of mTORC2-AKT1-PHLPP1. And I recommend adding descriptions of all molecules in a dedicated area.

Thank you for your valuable feedback on Figure 1. We acknowledge your concern regarding the distinction between positive and negative regulations, particularly within the depiction of mTORC2-AKT1-PHLPP1. In response to your suggestion, we have revised the legend to provide more comprehensive information about the type of regulation without altering the original image. Our aim is to maintain simplicity while ensuring clarity, and we believe that the updated legend achieves this balance effectively.

  1. Figure 2. I would recommend adding the major target proteins of CMA in each cellular process and their citations. It would be better if you organize these data within a table. My suggestion is that this information is not necessarily listed in a figure.

We fully understand your perspective, and your suggestion for a more detailed representation aligns with the approach taken in comprehensive reviews, such as Kaushik and Cuervo 2018. However, we would like to emphasize that our intention with this figure was to provide a concise overview of CMA, summarizing briefly its biological functions to offer a general perspective.

While we acknowledge the potential for a more detailed inclusion of CMA substrates and references in a table, we believe that such an approach might diverge from the specific goals and scope of our minireview. We aimed to maintain brevity and focus on providing a brief yet informative snapshot of CMA.

We truly thank for all Reviewers feedbacks and comments, we sincerely believe that the modifications added after all the inputs contribute to enhance the clarity and value of our minireview.

Reviewer 2 Report

Comments and Suggestions for Authors

The article titled "The Role of Chaperone-Mediated Autophagy in Tissue Homeostasis and Disease Pathogenesis" by Rut Valdor and Marta Martinez-Vicente, delves into the significance of chaperone-mediated autophagy (CMA) in lysosomal protein degradation. It emphasizes the importance of CMA in maintaining cellular homeostasis across various tissues and its involvement in the development of diseases related to aging, cancer, and disorders of the nervous and immune systems. The paper serves as a review, updating the current understanding of CMA's role in tissue homeostasis and disease pathogenesis, and highlighting its potential as a target for future disease treatments.

To assess the paper's merit for publication and its novelty compared to existing literature, several aspects should be considered:

  • Clarity and Structure: Evaluate if the article is well-organized, with clear headings, logical flow of content, and effective use of figures and tables.
  • Scientific Rigor and Accuracy: Assess the validity of the methodologies, the accuracy of the results, and whether the conclusions are supported by the data.
  • Literature Review and Novelty: Determine how well the article integrates existing literature on the subject and whether it offers new insights or data that significantly advance the understanding of the topic.
  •                 Discussion
  • Recent findings, such as those reported in  PMID: 37373581, have highlighted the significance of the tumor microenvironment in the progression of cancers such as clear-cell renal cell carcinoma, suggesting a potential link with chaperone-mediated autophagy mechanisms. Please include and discuss this interesting aspect. 
  • Recent studies, such as those reported in PMID: 37685983, have underscored the critical role of cancer stem cells (CSCs) in clear-cell renal cell carcinoma (ccRCC). These findings highlight the potential interactions between CSCs and chaperone-mediated autophagy (CMA) pathways. CSCs, characterized by their clonogenic ability, expression of stem cell markers, and tumorigenicity, may be influenced by CMA processes, which are pivotal in cellular homeostasis and degradation of proteins. The interplay between CSCs and CMA could offer new insights into the mechanisms underlying cancer progression and therapy resistance, emphasizing the importance of targeting these pathways in ccRCC treatment strategies. This integration of CSC dynamics with CMA pathways enriches our understanding of their combined roles in cancer pathogenesis and potential therapeutic interventions.
  • Significance and Impact: Consider the potential impact of the findings on the field and their relevance to current challenges or questions in the area of chaperone-mediated autophagy.
  • Writing Quality and Style: Examine the writing for clarity, conciseness, and technical precision. Ensure that the language is accessible to a broad scientific audience.
  • References: Check if the references are current, relevant, and appropriately cited, supporting the article's claims and situating it within the broader scientific discourse.
  • Ethical Considerations: Ensure that any ethical concerns, particularly if the study involves human or animal subjects, are adequately addressed and compliant with standard research ethics.

By focusing on these areas, a comprehensive evaluation of the article's suitability for publication can be made, taking into account its contribution to the existing body of knowledge.

Author Response

Thank you for considering the revision of our manuscript. We have addressed each concern raised by the reviewers as listed below:

REVIEWER 2:

Comments and Suggestions for Authors

The article titled "The Role of Chaperone-Mediated Autophagy in Tissue Homeostasis and Disease Pathogenesis" by Rut Valdor and Marta Martinez-Vicente, delves into the significance of chaperone-mediated autophagy (CMA) in lysosomal protein degradation. It emphasizes the importance of CMA in maintaining cellular homeostasis across various tissues and its involvement in the development of diseases related to aging, cancer, and disorders of the nervous and immune systems. The paper serves as a review, updating the current understanding of CMA's role in tissue homeostasis and disease pathogenesis, and highlighting its potential as a target for future disease treatments.

To assess the paper's merit for publication and its novelty compared to existing literature, several aspects should be considered:

  •  Clarity and Structure: Evaluate if the article is well-organized, with clear headings, logical flow of content, and effective use of figures and tables.
  •  Scientific Rigor and Accuracy: Assess the validity of the methodologies, the accuracy of the results, and whether the conclusions are supported by the data.
  •  Literature Review and Novelty: Determine how well the article integrates existing literature on the subject and whether it offers new insights or data that significantly advance the understanding of the topic.
  •                 Discussion
  • Recent findings, such as those reported in  PMID: 37373581, have highlighted the significance of the tumor microenvironment in the progression of cancers such as clear-cell renal cell carcinoma, suggesting a potential link with chaperone-mediated autophagy mechanisms. Please include and discuss this interesting aspect. 
  • Recent studies, such as those reported in PMID: 37685983, have underscored the critical role of cancer stem cells (CSCs) in clear-cell renal cell carcinoma (ccRCC). These findings highlight the potential interactions between CSCs and chaperone-mediated autophagy (CMA) pathways. CSCs, characterized by their clonogenic ability, expression of stem cell markers, and tumorigenicity, may be influenced by CMA processes, which are pivotal in cellular homeostasis and degradation of proteins. The interplay between CSCs and CMA could offer new insights into the mechanisms underlying cancer progression and therapy resistance, emphasizing the importance of targeting these pathways in ccRCC treatment strategies. This integration of CSC dynamics with CMA pathways enriches our understanding of their combined roles in cancer pathogenesis and potential therapeutic interventions.
  •  Significance and Impact: Consider the potential impact of the findings on the field and their relevance to current challenges or questions in the area of chaperone-mediated autophagy.
  •  Writing Quality and Style: Examine the writing for clarity, conciseness, and technical precision. Ensure that the language is accessible to a broad scientific audience.
  •  References: Check if the references are current, relevant, and appropriately cited, supporting the article's claims and situating it within the broader scientific discourse.
  •  Ethical Considerations: Ensure that any ethical concerns, particularly if the study involves human or animal subjects, are adequately addressed and compliant with standard research ethics.

By focusing on these areas, a comprehensive evaluation of the article's suitability for publication can be made, taking into account its contribution to the existing body of knowledge.

 Thank you for your thorough evaluation of our minireview. We appreciate your constructive feedback and we have tried to address each aspect you've highlighted, including integrating recent findings on the tumor microenvironment and cancer stem cells in clear-cell renal cell carcinoma (Xiao S., Xu G., Wang Z., Chong T. Chaperone-mediated autophagy can promote proliferation and invasion of renal carcinoma cells and inhibit apoptosis through PKM2. Oncol. Rep. 2021;46:214. doi: 10.3892/or.2021.8165). Your insights are invaluable, and we look forward to enhancing the manuscript based on your suggestions.

We truly thank for all Reviewers feedbacks and comments, we sincerely believe that the modifications added after all the inputs contribute to enhance the clarity and value of our minireview.

Round 2

Reviewer 2 Report

Comments and Suggestions for Authors

Round 2 Review Comment:

First and foremost, I would like to thank you for submitting a revised version of your manuscript titled "The Role of Chaperone-Mediated Autophagy in Tissue Homeostasis and Disease Pathogenesis". I appreciate the efforts made to improve the paper in response to my previous comments.

However, I must express some reservations regarding the current revision. While I acknowledge that the paper has been partially improved, I am not fully satisfied with the response provided, as it does not seem to comprehensively and specifically address all the issues raised earlier.

In particular, I wish to highlight that, although you have incorporated considerations on the role of the tumor microenvironment and cancer stem cells in clear-cell renal cell carcinoma, the specific citation I had suggested (PMID: 37373581) has not been included. I believe that the addition of this citation could significantly enrich the focus and depth of the discussion, adding substantial value to the paper.

Furthermore, I would like to renew my recommendation to integrate the manuscript with information from the article mentioned in my initial comment (PMID: 37685983). This contribution represents a crucial aspect for a thorough understanding of the role of CMA in the dynamics of cancer stem cells and clear-cell renal cell carcinoma, and its inclusion could significantly enhance the quality and relevance of your work.

If you disagree with any of the suggested revisions, please provide specific reasons for not incorporating them. Understanding your rationale is crucial for the review process and will assist in the manuscript's evaluation.

In conclusion, while appreciating the efforts made, I urge the authors to consider these additional improvements to ensure that the manuscript fully reflects the most recent and relevant findings in the field of CMA biology and its role in complex pathologies like cancer.

Author Response

We understand the importance of providing a well-rounded literature review, and we are committed to enhancing the manuscript's quality ensuring that it accurately reflects the current state of knowledge in CMA-related research. We have carefully considered your suggestion to incorporate these two additional references in the revised manuscript. After a thorough examination of the proposed articles, we would like to provide specific reasons for not including them: upon closer inspection, we observed that the suggested references, PMID 37685983 and PMID: 37373581, do not align with the primary focus of our work, which revolves around "the Role of Chaperone-Mediated Autophagy in Tissue Homeostasis and Disease Pathogenesis"​, while the mentioned articles discuss the role of macroautophagy not chaperone-mediated autophagy, which are obviously different autophagy pathways. Thus the thematic connection of these 2 papers to CMA is not evident. Furthermore, both suggested references are comprehensive reviews rather than experimental studies. We believe it is crucial to prioritize experimental research articles that directly contribute to the empirical foundation of CMA investigation.